# Effects of amino acid value-added urea on rice growth and nitrogen utilization

**Lin Cheng**[ID]*, **Zongya Wang**[⊕], **Shuangshuang Huang**[⊕], **Hongyan Wu**[⊕], **Ruichao Li**[⊕]

Department of Hydraulic Engineering, Wanjiang University of Technology, Maanshan, Anhui, China

⊕ These authors contributed equally to this work.
* 1550708563@qq.com

**Data Availability Statement:** All relevant data are within the manuscript and its Supporting Information files.

**Funding:** This study was funded by a key project of the Education Department of Anhui Province, China

## Abstract

To investigate the impact of amino acid value-added urea on rice growth and nitrogen utilization, this study aimed to provide insights into enhancing the quality and efficiency of traditional nitrogen fertilizers using small molecule active substances. Amino acids were added at 5‰ and 5% levels to create different levels of amino acid value-added urea (AU0.5 and AU5) by blending with urea as test materials. Pot experiments were conducted using 'Liangyouhua 6' rice as the test crop, with four treatment groups: non-urea (CK), regular urea (U), and amino acid value-added urea (AU0.5, AU5) at two different addition ratios. All treatments, except the control, had the same application rates of nitrogen, phosphorus, and potassium. After harvesting the rice, plant and soil samples were collected from various depths to analyze the nutrient composition of rice, nitrogen content of fertilizer in different soil layers, and $^{15}$N abundance. The results showed that amino acid-added urea significantly enhanced biomass accumulation in different parts of rice. Compared to U, rice straw and grain biomass increased by 25.27% to 32.74% and 21.71% to 27.77% under AU0.5 and AU5 treatments, respectively. In terms of nitrogen application, effective panicle and grain numbers per panicle in AU0.5 and AU5 rice were 17.37% to 21.05% and 8.76% to 15.33% higher than in U, with a significant difference between AU0.5 and U. Furthermore, total aboveground nitrogen and fertilizer nitrogen accumulation in rice treated with AU0.5 and AU5 increased by 3.59% to 5.09% and 3.31% to 8.49%, respectively, compared to U. The accumulation of fertilizer nitrogen in grains and leaves also showed increases of 2.86% to 6.32% and 4.38% to 16.25%, respectively, compared to U. This study found that the application of amino acid value-added urea had a significant impact on the accumulation of fertilizer nitrogen in straw. Further analysis showed that it improved both the apparent nitrogen utilization efficiency and fertilizer nitrogen utilization efficiency. Compared to ordinary urea, the apparent nitrogen utilization efficiency of AU0.5 and AU5 increased by 23.71% and 33.93%, respectively, while the utilization efficiency of $^{15}$N increased by 15.66% and 6.78%, respectively. The residual fertilizer nitrogen in soil treated with amino acid value-added urea was mainly found in the 0-30cm soil layer, reducing nitrogen leaching downwards. Additionally, the nitrogen loss rate was significantly lower (reduced by 12.39%-12.97%) compared to regular urea. The difference between AU0.5 and AU5 was not significant, but AU5 showed an 18.80% higher fertilizer nitrogen residual rate than U. Overall, the study concluded that amino acid value-added urea promoted rice growth by enhancing nitrogen absorption,

(grant numbers:2022AH052438).CHENG Lin planned and designed the research.

**Competing interests:** The authors have declared that no competing interests exist.

improving transport to grains, increasing nitrogen fertilizer efficiency, reducing nitrogen leaching, and lowering nitrogen loss rate. The best results were observed with the addition of 5‰ amino acid.

# 1 Introduction

Rice, as one of the three major food crops, plays a crucial role in China's food security [1]. Urea, the most commonly used nitrogen fertilizer in Chinese agricultural production, accounts for 70% of total nitrogen fertilizer consumption, surpassing international safety standards [2,3]. However, due to its strong activity and multiple loss pathways, only a small portion of urea can be absorbed by crops and fixed by the soil, with the majority entering water and the atmosphere through various processes. Reports indicate that in China, the utilization rate, residual rate, and loss rate of urea nitrogen are 40.70%, 19.30%, and 40.00% [4] respectively, leading to resource wastage and impacting ecological security [5–7]. Developing value-added urea, primarily through the addition of biological macromolecular substances like humic acid and alginate, is key to improving fertilizer efficiency, reducing nitrogen loss, and enhancing environmental benefits. While current research on the synergistic mechanism of these composite substances is complex, small molecule active substances, derived from natural organic compounds, offer a promising solution. These substances, sourced from a variety of safe and environmentally friendly sources, contain functional groups that can undergo complex reactions with urea, ultimately enhancing the quality and efficiency of traditional urea.

Amino acid-based small molecule active substances, categorized as plant stimulants, play a crucial role in enhancing plant growth and metabolism, regulating physiological development, and are commonly utilized as fertilizer enhancers [8–10]. Studies have shown that the application of amino acids through foliar spraying not only improves stress conditions in strawberry cultivation but also boosts fruit yield [11]. For instance, Wang Bei's research demonstrated that foliar spraying with amino acid-containing water-soluble fertilizer significantly increased the yield of chili peppers and kidney beans while enhancing the soil microbial environment [12]. Additionally, spraying amino acids on mung beans led to notable improvements in root activity, leaf number, biomass, and chlorophyll content, with a positive correlation to the spraying concentration [13]. Despite these findings, existing studies primarily focus on the impact of amino acids as additives in water-soluble fertilizers on crop yield, quality, and nutrient utilization efficiency. Limited research exists on the value-added urea derived from amino acids and urea, particularly regarding the effects of different concentrations of amino acids and urea fusion on fertilizers, crops, soil, and their subsequent application outcomes. Therefore, investigating the effects of chelated urea with varying levels of amino acid addition on nitrogen transformation characteristics is of significant theoretical and practical importance in elucidating the mechanisms by which amino acid-enhanced urea promotes rice growth, nutrient accumulation, and utilization.

Based on previous research, this experiment utilized 5 ‰ and 5% amino acids to chelate with urea in order to produce varying concentrations of amino acid value-added urea. Through the use of rice pot culture and 15N tracking technology, the study examined the impact of amino acid value-added urea on rice yield, nitrogen utilization, fertilizer nitrogen residue, and soil nitrogen loss. Additionally, the research aimed to clarify the effects of amino acid value-added urea on crop growth and nutrient accumulation, providing valuable data and a theoretical foundation for enhancing fertilizer performance, improving fertilizer utilization efficiency, and developing novel fertilizers containing small molecule active substances.

## 2 Materials and methods

### 2.1 Test materials

**2.1.1 Test fertilizer.** (1) Amino acid synergist was developed by the Institute of Resources and Agricultural Regionalization of the Chinese Academy of Agricultural Sciences. It contains a total amino acid content of 20%, predominantly made up of glutamic acid, lysine, valine, and alanine.

(2)Amino acid synergists are added to urea with a $^{15}$N abundance of 10.05% in a ratio of 5 ‰ and 5%, respectively. The mixture is fully melted at 130˚C, cooled, crushed, and sieved to produce AU0.5 (5 ‰ amino acid value-added urea) and AU5 (5% amino acid value-added urea).

(3)$^{15}$N urea (U): The experiment was repeated using the same urea as in step 2, but this time without adding an amino acid synergist. The urea was melted, cooled, crushed, and sieved to obtain $^{15}$N urea (U), which was used as the control for step 2.

Phosphate fertilizer and potassium fertilizer are commercially available brands of potassium dihydrogen phosphate and potassium chloride, and the basic properties of urea used are shown in Table 1.

**2.1.2 Test soil.** Soil samples were collected from the 0–30 cm layer of Hefei Agricultural Park, Anhui Agricultural University. The samples were processed by removing stones and roots, followed by air drying and grinding through a 2 mm sieve. The soil type was identified as paddy soil. The soil's basic physical and chemical properties included 10.24 g·kg$^{-1}$ of organic matter, 0.78 g·kg$^{-1}$ of total nitrogen, 42.4 mg·kg$^{-1}$ of alkaline nitrogen, 9.35 mg·kg$^{-1}$ of available phosphorus, 138.62 mg·kg$^{-1}$ of available potassium, and a pH of 6.57.

**2.1.3 Test crop.** Rice: Variety 'Liangyouhua 6' (*Oryza sativa L.*).

### 2.2 Experimental design

The rice pot experiment was conducted at the experimental field of Wanjiang University of Technology from May to September 2021. Four treatments were established: (1) control with no nitrogen fertilizer (CK), (2) 15N urea (U), (3) 5 ‰ amino acid value-added urea (AU0.5), and (4) 5% amino acid value-added urea (AU5). Based on an average soil bulk density of 1.36 g/cm3, each test pot (100 cm high, 25 cm inside diameter) contained 55 kg of soil. Rice seedlings were transplanted into each pot with 5 holes, each hole containing 5 plants. The fertilization levels of N, $P_2O_5$, and $K_2O$ were 0.15, 0.2, and 0.2 g·kg$^{-1}$, respectively (fertilizer amounts calculated on a dry soil basis). All fertilizers were applied once before rice transplanting and thoroughly mixed into the 0–30 cm soil layer. Each treatment was replicated 5 times and randomly arranged. Throughout the rice growth cycle, management practices followed local conventional cultivation techniques. Rice was harvested on September 30, 2021.

### 2.3 Sample

After the rice reaches maturity, three representative rice panicles are selected from each pot. These panicles are cleaned with flowing water, and the aboveground plants are divided into

**Table 1. Amino acid addition ratio and nitrogen content of experimental urea.**

| Treatment | Amino acid(%) | Total N(%) | $^{15}$N abundance(%) |
|:---:|:---:|:---:|:---:|
| U | 0 | 45.85 | 10.05 |
| AU0.5 | 0.5 | 45.44 | 10.01 |
| AU 5 | 5 | 43.52 | 10.00 |

three parts: stem, leaf, and spike, which are then placed into sample bags. The samples are dried at 105°C for 30 minutes, followed by drying at 65°C until a constant weight is achieved. The dry weight of each part is then measured after drying. The weighed samples are crushed through a 0.178 mm sieve for further analysis. Soil samples were collected post rice harvest, with samples taken from 's' type layers (0~30 cm, 30~60 cm, 60~90 cm) in each pot using a 100 ml ring knife. Each layer was sampled three times in parallel. Care was taken to remove any roots or stones from the samples, which were then dried, ground, and screened for subsequent analysis.

### 2.4 Determination method

Rice biomass is calculated as the average of three replicated experiments. The total nitrogen content of plants and various soil layers is analyzed using the Kjeldahl method [14], while the $^{15}$N abundance is determined using the Delta V advanced isotope mass spectrometer from Elementar, Germany [15].

### 2.5 Data processing

After the data were sorted by Excel 2017, the following indicators were calculated according to references [16–18], and SPSS 22 and Duncan were used for variance analysis and correlation analysis.

Plant total nitrogen accumulation (g pot$^{-1}$) = plant total nitrogen content × Plant biomass

Plant fertilizer nitrogen accumulation (g/pot) = plant total nitrogen accumulation × ($^{15}$N atomic percentage of plant exceeds/$^{15}$N atomic percentage of fertilizer exceeds)

Residual fertilizer nitrogen in each soil layer (g/pot) = total nitrogen content in each soil layer × Dry weight of each soil layer × Each soil layer ($^{15}$N atomic percent of soil/$^{15}$N atomic percent of fertilizer)

Apparent nitrogen use efficiency (%) = (total nitrogen content of plant applying nitrogen fertilizer × Plant biomass—total nitrogen content of plants without nitrogen fertilizer × Plant biomass)/nitrogen application rate × one hundred

Fertilizer nitrogen utilization rate (%) = plant fertilizer nitrogen accumulation/fertilizer nitrogen application rate × one hundred

Fertilizer nitrogen residue rate (%) = fertilizer nitrogen residue in soil/fertilizer nitrogen application rate × one hundred

Fertilizer nitrogen loss rate (%) = 100%- (total accumulation of fertilizer nitrogen in plant +total residue of fertilizer nitrogen in soil)/fertilizer nitrogen application rate × 100%

## 3 Result and analysis

### 3.1 Effects of amino acid added urea on biomass and yield components of rice

As shown in Fig 1, the application of nitrogen fertilizer significantly enhanced the growth of rice, with varying effects observed based on the type of nitrogen fertilizer used. Both AU0.5 and AU5 treatments led to a substantial increase in total aboveground biomass, ranging from 19.89% to 25.81% compared to ordinary urea (U). There was no significant difference between the effects of AU0.5 and AU5 treatments. Furthermore, AU0.5 and AU5 significantly boosted the biomass of rice grain and straw by 21.71% to 27.77% and 25.27% to 32.74%, respectively, when compared to U. However, the impact on rice leaf biomass increase was not statistically significant. An in-depth analysis of the influence of each treatment on rice yield components is presented in Table 2. Nitrogen application, in general, resulted in a rise in panicle length and

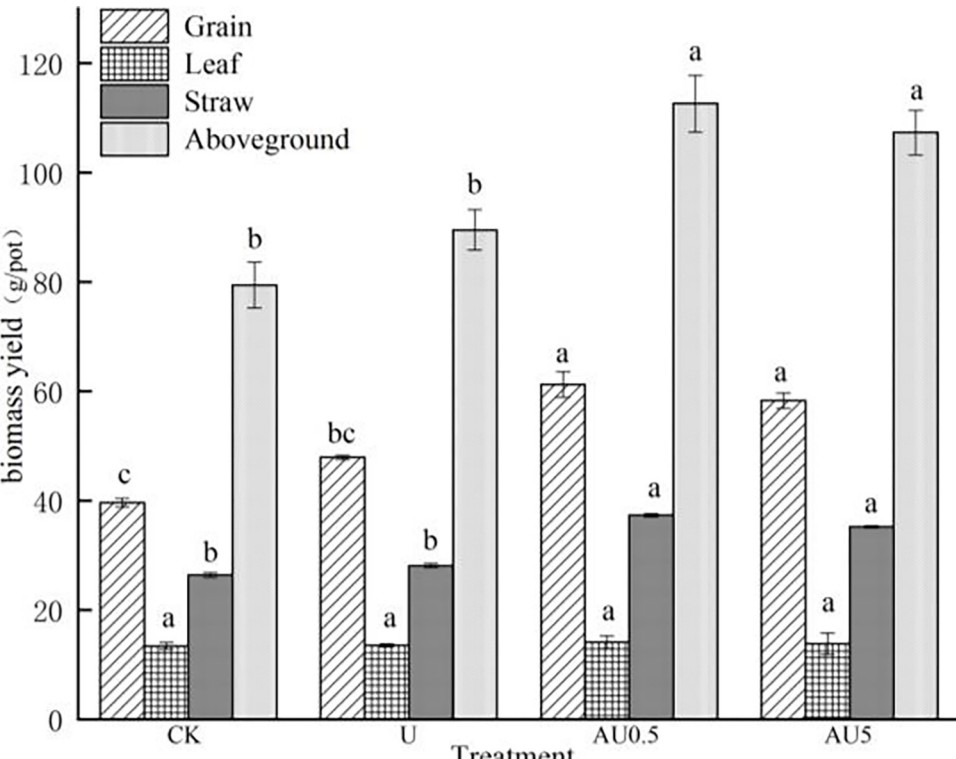

**Fig 1. Effect of amino acid value-added urea on biomass yield of rice.**

grain number per panicle by 7.28% to 16.40% and 9.60% to 26.40%, respectively, compared to no nitrogen application. Moreover, the effective panicles and grains per panicle of rice treated with AU0.5 and AU5 increased by 17.37% to 21.05% and 8.76% to 15.33%, respectively, in comparison to U, with a significant difference noted between AU0.5 and U. Additionally, the application of amino acid-enriched urea showed a slight improvement in panicle length of rice, though not significantly different from ordinary urea.

## 3.2 Effect of amino acid added urea on nitrogen uptake by rice

The impact of various treatments on the total nitrogen accumulation in rice shoots is illustrated in Fig 2A. In comparison to standard urea, the total nitrogen accumulation in rice shoots increased by 3.59% to 5.09% with amino acid value-added urea treatment, with no significant variance observed among the different nitrogen treatments. Notably, the AU0.5 and AU5 treatments notably boosted the total nitrogen accumulation in rice leaves by 20.61% and 11.59% respectively compared to U, while the nitrogen accumulation in grains and stems did

**Table 2. Effects of amino acid value-added urea on rice grain yield components.**

| Treatment | Spike length(cm) | Spike number(No./pot) | Number of grains(No./Spike) | Seed-setting rate(%) | Thousand grain weight(g) |
|---|---|---|---|---|---|
| CK | 15.67±1.2b | 13.5±0.6c | 125±13.3c | 90.92±5.1a | 24.40±0.22a |
| U | 16.81±1.7ab | 19.0±2.0b | 137±2.5bc | 91.08±1.9a | 25.57±0.41a |
| AU0.5 | 17.52±1.0ab | 23.0±1.7a | 158±13.1a | 92.37±2.4a | 25.31±0.38a |
| AU5 | 18.24±2.0a | 22.3±1.0ab | 149±4.2ab | 91.75±4.1a | 25.87±0.32a |

Note: Different letters in a column mean significant difference among treatments at the 5% level. (The same below).

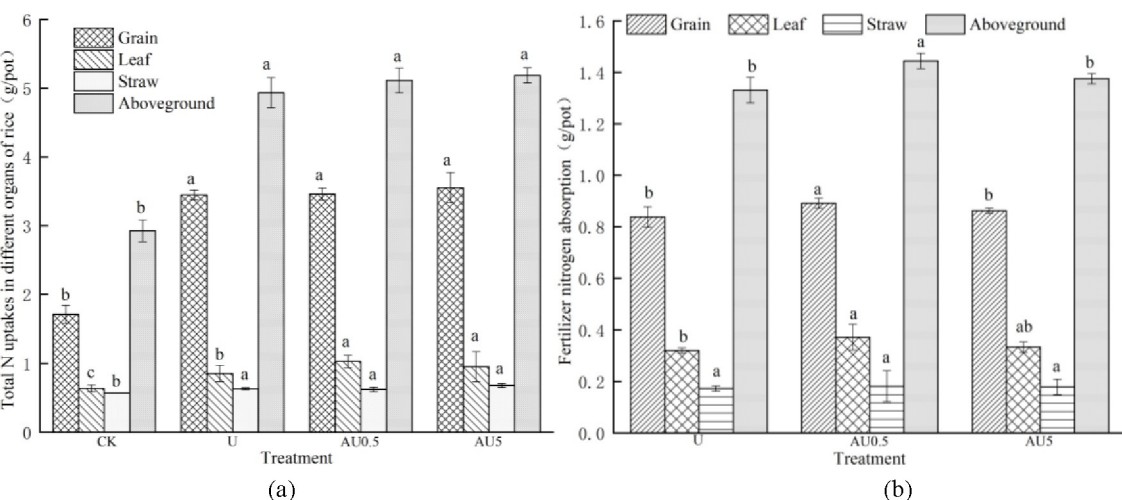

**Fig 2. Effect of amino acid added urea on nitrogen accumulation and fertilizer nitrogen absorption in rice.**

not reach significant levels. Further examination of the nitrogen accumulation in rice aboveground due to fertilizer application is presented in Fig 2B. The utilization of amino acid value-added urea resulted in an increase in the accumulation of fertilizer nitrogen in rice aboveground. Specifically, under AU0.5 treatment, the total accumulation of fertilizer nitrogen in rice aboveground increased by 8.49% compared to U, a significant difference, and by 3.31% under AU5 treatment, a non-significant difference. Analysis of the fertilizer nitrogen accumulation in different parts of rice revealed that the accumulated fertilizer nitrogen was primarily concentrated in grains, accounting for 61.70% to 62.96% of the total accumulated fertilizer nitrogen in the upper part of the plant. In comparison to U, the accumulation of fertilizer nitrogen in grains and leaves of rice under AU0.5 and AU5 treatments increased by 2.86% to 6.32% and 4.38% to 16.25% respectively, with a significant difference observed only between AU0.5 and U treatments. However, the different nitrogen treatments did not have a significant impact on the accumulation of fertilizer nitrogen in rice stems.

The data presented in Table 3 indicates that approximately 26.51% to 28.25% of the nitrogen accumulated in rice is derived from fertilizer nitrogen. Interestingly, the fertilizer nitrogen accumulation under the AU0.5 treatment contributes the most to the total nitrogen accumulation, significantly surpassing other treatments. It is worth noting that there were no significant variations observed in the proportion of fertilizer nitrogen to total nitrogen accumulation in leaves and stems across different nitrogen treatments.

### 3.3 Effect of amino acid added urea on nitrogen use efficiency of rice

The utilization rate of rice for different types of nitrogen fertilizer is presented in Fig 3. When comparing ordinary urea with amino acid value-added urea, it is evident that the utilization rate of nitrogen fertilizer for rice significantly improves. The apparent nitrogen use efficiency

**Table 3. Proportion of fertilizer nitrogen absorbed by each part of rice in nitrogen accumulation (%).**

| Treatment | Grain | Leaf | Straw | Aboveground |
|---|---|---|---|---|
| U | 24.30±0.44b | 37.47±0.09a | 27.33±0.82a | 26.97±1.0b |
| AU0.5 | 25.75±0.38a | 36.12±0.07a | 29.10±1.2a | 28.25±0.3a |
| AU5 | 24.26±0.32b | 35.05±0.22a | 26.32±1.7a | 26.51±0.8b |

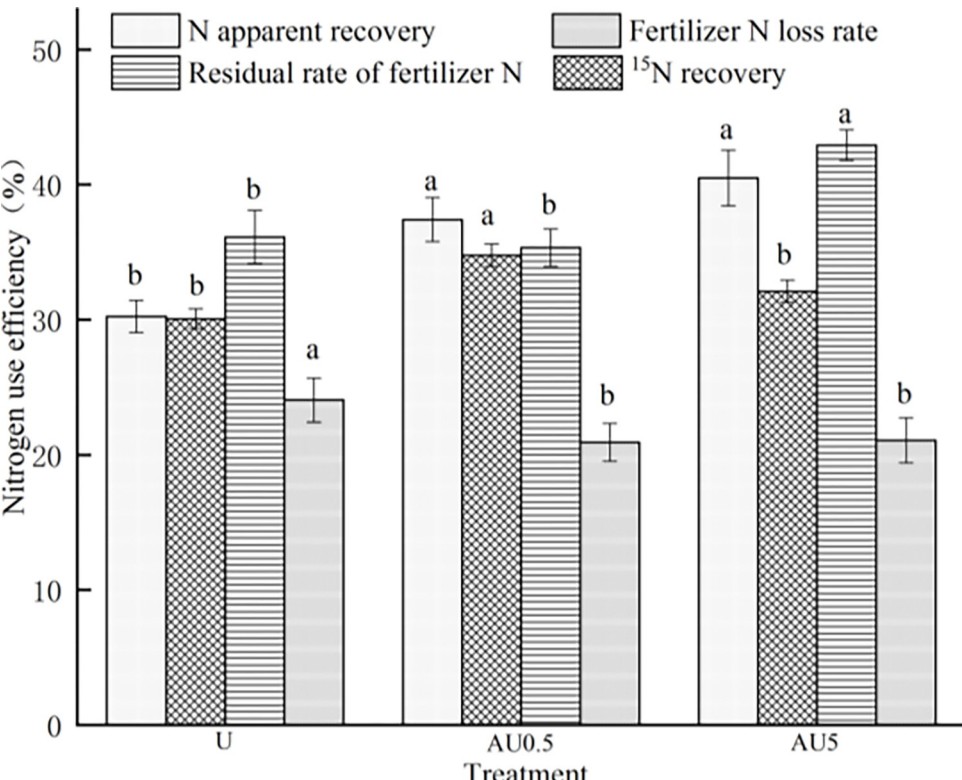

**Fig 3. Effects of amino acid value-added urea on rice nitrogen use efficiencies.**

of AU0.5 and AU5 treatments was 23.71% and 33.93% higher, respectively, than that of the U treatment. The 15N utilization rates of AU0.5 and AU5 treatments were 15.66% and 6.78% higher than those of the U treatment, with only the 15N utilization rate of AU0.5 treatment and U treatment reaching significance. Further analysis on the fate of fertilizer nitrogen revealed that, aside from crop accumulation, 35.34%-42.91% of fertilizer nitrogen remained in the soil, while 20.93%-24.05% was lost through ammonia volatilization and leaching. The residual rate of fertilizer nitrogen increased by 18.80% under AU5 treatment compared to U, with a significant difference. No significant difference was observed in the residual rate of fertilizer nitrogen between AU0.5 treatment and U. Moreover, the loss rate of nitrogen fertilizer was reduced by 12.39%-12.97% under amino acid added urea treatment compared to ordinary urea, with a significant difference. There was no significant difference in the loss rate of nitrogen fertilizer among different amino acid additions.

### 3.4 Effect of amino acid added urea on distribution of fertilizer nitrogen in soil profile

The analysis of residual fertilizer nitrogen distribution in the soil post rice harvest (Table 4) reveals that the majority of residual fertilizer nitrogen is found in the 0–60 cm soil layer across

**Table 4. Effects of amino acid value-added urea on fertilizer N distributions in soil profiles (g pot$^{-1}$).**

| Soil layer (cm) | U | AU0.5 | AU5 |
| --- | --- | --- | --- |
| 0~30 | 0.253±0.001b | 0.249±0.002b | 0.292±0.002a |
| 30~60 | 0.075±0.002b | 0.078±0.001b | 0.09±0.002a |
| 60~90 | 0.003±0.001a | 0.004±0.001a | 0.004±0.001a |

various nitrogen treatments. Specifically, the 0–60 cm soil layer contains over 98% of the total residual fertilizer nitrogen in the 0–90 cm soil layer for treatments U, AU0.5, and AU5. Within this range, the AU5 treatment exhibited the highest residual nitrogen levels in both the 0–30 cm and 30–60 cm soil layers, showing a significant increase of 15.42% and 20.00% compared to treatment U. However, there was no significant difference in residual fertilizer nitrogen levels in the 60–90 cm soil layer among the different nitrogen treatments.

## 4 Discussion

### 4.1 Effects of amino acid added urea on rice biomass and yield components

Numerous studies have demonstrated that the single application of amino acids or their combined application with urea can enhance crop yield [18–21]. For instance, spraying a specific concentration of amino acids during wheat flowering has been shown to increase the seed setting rate per spike, improve yield, enhance quality, and optimize fertilizer utilization [22]. Additionally, the combined application of amino acids and fertilizers has been found to stimulate rice seedling growth, boost yield, and enhance the soil biological environment [23–25]. In this study, amino acid-enriched urea was prepared by incorporating amino acids into urea. Compared to regular urea, this enriched version significantly increased rice aboveground biomass and grain yield, aligning with previous research findings and suggesting a positive impact on yield. Analysis of rice yield composition revealed that the application of amino acid-enriched urea notably augmented the number of effective panicles and grains per panicle, indicating that the enhancement in yield is primarily attributed to an increase in rice panicles, consistent with Yuan's research findings [18].

### 4.2 Effect of amino acid added urea on nitrogen uptake and utilization by rice

Amino acid-added urea can enhance the absorption of fertilizer nitrogen by rice and improve the nitrogen fertilizer utilization rate. Research findings demonstrate that adding 5‰ amino acid to urea significantly boosts the urea nitrogen utilization rate in rice. This enhancement is attributed to the ability of amino acid-added urea to facilitate nitrogen accumulation and assimilation in rice during later stages, leading to increased nitrogen accumulation in rice, aligning with Cheng's previous research results [25]. However, as the amount of added amino acids increases, the nitrogen use efficiency of rice decreases, primarily due to the higher residual amount of fertilizer nitrogen in the soil resulting from the application of 5% amino acid-added urea. This decline may be linked to the production of more stable complex products post-melt granulation of 5% amino acid-added urea, enhancing its slow-release property. Additionally, variations in rice varieties, planting density, climate, and other factors may also play a role [26]. Amino acid-added urea can raise the total accumulation of fertilizer nitrogen in rice plants by boosting the accumulation of fertilizer nitrogen in rice grains. This suggests that amino acid-added urea can improve crop nitrogen absorption and utilization by directly participating in plant growth and development regulation as important biological stimulants. This involvement leads to enhanced crop root growth, increased expression of related enzyme genes, improved nutrient element accumulation in roots, and ultimately, enhanced nitrogen accumulation efficiency [25–28]. The interaction between amino acids and urea can result in complexation, chelation, or physical-chemical adsorption. This can delay the dissolution and transformation time of nitrogen fertilizer, alter the release properties of urea, and align nutrient release with the demands of rice growth [25,29,30]. Moreover, the introduction of extra carbon sources may impact soil enzyme activity and carbon-nitrogen ratio, enhancing soil mineralization and providing additional nitrogen for rice cultivation [31].

### 4.3 Effect of amino acid added urea on distribution, residue and loss of fertilizer nitrogen in soil

By examining the distribution and residue of amino acid-enriched urea on nitrogen fertilizer in soil, the findings indicated that this type of urea not only boosted the overall accumulation of fertilizer nitrogen in crops, but also increased the nitrogen residue in soil while reducing the nitrogen fertilizer loss rate. One possible reason for this effect could be the reduction in soil urease activity and inhibition of urea transformation by amino acid-enriched urea, leading to decreased nitrogen loss from ammonia volatilization [25]. Additionally, the promotion of rice root growth by amino acid-enriched urea may enhance nutrient absorption, further lowering the loss rate of fertilizer nitrogen. Nevertheless, the precise theoretical mechanism requires further investigation. The study revealed that more than 60% of the residual nitrogen fertilizer in the 0–30 cm soil layer was found in the 0–90 cm soil layer, with significantly higher residual nitrogen levels in various soil layers under 5% amino acid-enriched urea treatment compared to standard urea treatment, suggesting that the addition of 5% amino acid to urea can effectively mitigate the risk of nitrogen leaching. While this research focused on a single soil type to assess the impact of amino acid-enriched urea on rice growth and nitrogen transformation, future studies should delve into the molecular structure characteristics of amino acid-enriched urea, the distinctions between fusion and physical mixing on urea transformation, and the application efficacy in diverse soil types.

## 5 Conclusions

Compared to conventional urea, the addition of amino acids to urea can enhance rice yield and improve nitrogen accumulation and utilization. However, the impact of different levels of amino acid additions (5‰ and 5%) on rice yield and the fate of fertilizer nitrogen varies.

(1) The application of amino acid enriched urea led to a yield increase of 10.07%-12.19%, primarily by boosting the number of effective panicles and grains per panicle in rice. No significant differences were observed among different levels of amino acid additions.

(2) Amino acid enriched urea, in comparison to conventional urea, can enhance nitrogen uptake in rice, facilitate nitrogen transport to the aboveground parts, and enhance the apparent nitrogen fertilizer utilization rate and fertilizer nitrogen utilization rate, with the best results seen with 5‰ amino acid enriched urea.

(3) The residual fertilizer nitrogen in soil treated with amino acid enriched urea was predominantly found in the 0-30cm soil layer, reducing the risk of nitrogen leaching and effectively preventing nitrogen loss, with the 5% amino acid enriched urea showing the most favorable outcomes.

## Supporting information

**S1 File. Diagram file.**
(7Z)

**S2 File. Minimal data set.**
(XLS)

## Author Contributions

**Data curation:** Lin Cheng, Zongya Wang.

**Formal analysis:** Lin Cheng, Shuangshuang Huang.

**Methodology:** Lin Cheng, Hongyan Wu, Ruichao Li.

**Project administration:** Lin Cheng.

**Writing – original draft:** Lin Cheng.

**Writing – review & editing:** Lin Cheng.

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
