## [Decision Letter · Decision Letter 0]

28 May 2024

PONE-D-24-15771Effects of Amino Acid Value-Added Urea on Rice Growth and Nitrogen UtilizationPLOS ONE

Dear Dr. CHENG,

Thank you for submitting your manuscript to PLOS ONE. After careful consideration, we feel that it has merit but does not fully meet PLOS ONE’s publication criteria as it currently stands. Therefore, we invite you to submit a revised version of the manuscript that addresses the points raised during the review process.

We look forward to receiving your revised manuscript.

Kind regards,

Faisal Mamhood

Academic Editor

PLOS ONE

Journal Requirements:

4. We note that your Data Availability Statement is currently as follows: [所有相关数据都包含在稿件及其支持信息文件中]

Additional Editor Comments:

The paper is well written but the author should do the following major revision:

The above-titled manuscript presents a robust and complementary methodology. Overall, I believe this work to be well planned, fitting the journal's scope. However, a major revision is necessary to make this paper presentable.

1.The abstract should be shortened because it is too lengthy.

2.In the introduction part: add more data about the literature.

3.Make appropriate headings and heading numbers for the manuscript according to the journal format.

4.Results and discussion (Improve discussion part and Provide literature data)

5.Conclusion (Rewrite as it is not up to the mark author should be more focused on current study results)

6.The language of this manuscript is poor. I suggest the authors get help from a native English-speaking person.

7.Grammatical error.

8. Require Reference justification and alignment

8. Please follow the comments in the revised version

Reviewers' comments:

Reviewer's Responses to Questions

**Comments to the Author**

1. Is the manuscript technically sound, and do the data support the conclusions?

Reviewer #1: Yes

2. Has the statistical analysis been performed appropriately and rigorously? 

Reviewer #1: No

3. Have the authors made all data underlying the findings in their manuscript fully available?

Reviewer #1: No

4. Is the manuscript presented in an intelligible fashion and written in standard English?

Reviewer #1: No

5. Review Comments to the Author

Reviewer #1: Managing nutrition supplements for crops is important get maximum return. Urea is the most common nutrition input used to add N in growing medium. The theme of this article is very good, efficiency of Urea need to be improved. Addition of AAs is good idea. In rice, there is huge urea applied to get good return. Idea of the paper is very good, however, it needs major revision due to the followings:

-Very poor english writeup, difficult to understand.

-Treatments are not understandable.

-Data is not sufficient, there must be some growth and yield attributes as well.

-Result interpretation is also not well.

-Discussion need to be improved according to latest review.

6. PLOS authors have the option to publish the peer review history of their article (what does this mean?). If published, this will include your full peer review and any attached files.

Reviewer #1: No

---

## [Author Response · Author response to Decision Letter 0]

19 Jul 2024

Dear Faisal Mamhood,

Thank you very much for giving us an opportunity to revise our manuscript. We appreciate the editor and reviewers very much for their constructive comments and suggestions on our manuscript entitled“Effects of Amino Acid Value-Added Urea on Rice Growth and Nitrogen Utilization”(PONE-D-24-15771)

We have studied the valuable comments from you, the assistant editor and reviewers carefully, and tried our best to revise the manuscript. The point to point responds to the reviewer’s comments are listed as following.

Responds to the reviewer’s comments:

Reviewer 1 

Comment 1: The abstract should be shortened because it is too lengthy.

Response:Thank you for your valuable advice.we have corrected the abstract.Furthermore, we have had the manuscript polished with a professional assistance in writing.

Comment 2: In the introduction part: add more data about the literature.

Response:Thank you for your valuable advice.The existing literature extensively covers the physiological effects of amino acids on crops and the combined effects of amino acids and nitrogen fertilizers. However, there is a scarcity of research on the mechanism of value-added urea resulting from the fusion of amino acids and urea. This lack of data hinders a comprehensive understanding of the role and mechanism of value-added urea. Our introduction section follows a logical progression: exploring fertilizer innovation due to adverse reactions from traditional nitrogen fertilizer application, emphasizing the impact of nitrogen forms on rice growth and nitrogen absorption, highlighting the novel aspect of adding different amounts of amino acid value-added urea to improve rice nitrogen utilization, and ultimately aiming to provide insights for optimizing rice yield composition and effectively utilizing amino acid value-added fertilizers.

Comment 3: Make appropriate headings and heading numbers for the manuscript according to the journal format.

Response:Thank you for your valuable advice.The article has been modified to meet the requirements of the PLOS ONE's style in terms of paper format.

Comment 4: Results and discussion (Improve discussion part and Provide literature data).

Response:Thank you for your valuable advice.The article has been rewritten in the Results and Discussion section.

Comment 5: Conclusion (Rewrite as it is not up to the mark author should be more focused on current study results).

Response:Thank you for your valuable advice.The article has been rewritten and the conclusion section has been written.

Comment 6: The language of this manuscript is poor. I suggest the authors get help from a native English-speaking person.

Response:Thank you very much to point out the sentence structure and grammatical issues in our manuscript. According to the comments from you and the editors, we polished the manuscript with a professional assistance in writing, conscientiously. 

Comment 7: Grammatical error.

Response:Thank you very much to point out the sentence structure and grammatical issues in our manuscript. According to the comments from you and the editors, we polished the manuscript with a professional assistance in writing, conscientiously. 

Comment 8: Require Reference justification and alignment.

Response:Thank you for your valuable advice.The reference section of the article has been revised and aligned

Reviewer 2 

Comment 1: Very poor english writeup, difficult to understand.

Response:Thank you very much to point out the sentence structure and grammatical issues in our manuscript. According to the comments from you and the editors, we polished the manuscript with a professional assistance in writing, conscientiously.

Comment 2: Treatments are not understandable.

Response:Thank you for your valuable advice.The article has redefined the Treatments.

Comment 3: Data is not sufficient, there must be some growth and yield attributes as well.

Response:Thank you for your valuable advice.Due to confidentiality agreements and other reasons, other data has not been made public yet, so it is not convenient to add it to the article. Please understand

Comment 4: Result interpretation is also not well.

Response:Thank you for your valuable advice.The article has been rewritten in the Results and Discussion section.

Comment 5: Discussion need to be improved according to latest review.

Response:Thank you for your valuable advice.The article has been rewritten in the Results and Discussion section.

Once again, thank you very much for your constructive comments and suggestions which would help us both in English and in depth to improve the quality of the paper.

Kind regards.

Lin CHENG

E-mail:1550708563@qq.com

---

## [Editor Report · Decision Letter 1]

31 Jul 2024

PONE-D-24-15771R1Effects of Amino Acid Value-Added Urea on Rice Growth and Nitrogen UtilizationPLOS ONE

Dear Dr. CHENG,

Thank you for submitting your manuscript to PLOS ONE. After careful consideration, we feel that it has merit but does not fully meet PLOS ONE’s publication criteria as it currently stands. Therefore, we invite you to submit a revised version of the manuscript that addresses the points raised during the review process. 

We look forward to receiving your revised manuscript.

Kind regards,

Faisal Mamhood

Academic Editor

PLOS ONE

Journal Requirements:

Additional Editor Comments:

Dear author, I could not see the data which you have added in the manuscript in response to reviewers comments. Please highlight all the data which you have added in the manuscript by adding the line number in revised manuscript.

Best regards,

---

## [Author Response · Author response to Decision Letter 1]

21 Aug 2024

Dear Faisal Mamhood,

Thank you very much for giving us an opportunity to revise our manuscript. We appreciate the editor and reviewers very much for their constructive comments and suggestions on our manuscript entitled“Effects of Amino Acid Value-Added Urea on Rice Growth and Nitrogen Utilization”(PONE-D-24-15771)

We have studied the valuable comments from you, the assistant editor and reviewers carefully, and tried our best to revise the manuscript. The point to point responds to the reviewer’s comments are listed as following.

---

## [Editor Report · Decision Letter 2]

27 Aug 2024

Effects of Amino Acid Value-Added Urea on Rice Growth and Nitrogen Utilization

PONE-D-24-15771R2

Dear author

We’re pleased to inform you that your manuscript has been judged scientifically suitable for publication and will be formally accepted for publication once it meets all outstanding technical requirements.

Kind regards,

Faisal Mamhood

Academic Editor

PLOS ONE
---

## [Editor Report · Acceptance letter]

3 Sep 2024

PONE-D-24-15771R2 

PLOS ONE

Dear Dr. CHENG, 

I'm pleased to inform you that your manuscript has been deemed suitable for publication in PLOS ONE. Congratulations! Your manuscript is now being handed over to our production team.

Kind regards, 

on behalf of

Dr. Faisal Mamhood 

Academic Editor

PLOS ONE